

# Ground-based noontime D-region electron density climatology over northern Norway

Toralf Renkwitz[1], Mani Sivakandan[1,2], Juliana Jaen[1], and Werner Singer[1]

[1]Leibniz-Institute of Atmospheric Physics at the Rostock University, Schloss-Str. 6, 18225 Kühlungsborn, Germany
[2]Faculty of Mathematics and Natural Sciences, University of Rostock, Rostock, Germany

**Correspondence:** Toralf Renkwitz
(renkwitz@iap-kborn.de)

**Abstract.**

The bottom part of the earth's ionosphere is the so-called D-region, which is typically less intense than the upper regions. Despite the comparably lower electron number density, the ionization state of the D-region has a significant influence on signal absorption for propagating lower to medium radio frequencies. We present local noon climatologies of electron number density in the middle atmosphere at high latitudes as observed by an active radar experiment. The radar measurements cover nine years from the solar maximum of cycle 24 to the beginning of cycle 25. Reliable electron densities are derived by employing signal processing, applying interferometry methods, and the Faraday International Reference Ionosphere (FIRI) model. For all years a consistent spring-autumn asymmetry of the electron number density pattern as well as a sharp decrease at the beginning of October was found. These findings are consistent with VLF studies showing equivalent signatures for nearby propagation paths. It has been suggested that the meridional circulation associated with downwelling in winter could cause enhanced electron densities through NO transport. However, this mechanism lacks to explain the reduction in electron density in early October.

## 1 Introduction

The lower part of the ionosphere is called the D-region and generally refers to altitudes below $100\,\mathrm{km}$ (Mitra, 1968). The ionization of the nitric oxide (NO) and meta-stable oxygen molecules ($O_2$) in this region's electron concentration is mainly controlled by the Lyman Alpha line and UV radiation, respectively. This is also a region where cluster ions are observed. The D-region also co-exists with the mesosphere which is the most dynamic region in the earth's atmosphere, where temperature decreases with altitude and most of the wave breaking occurs. Diurnal variation of the D-region electron density is controlled by the changes in the solar zenith angle. Unlike the E- and F-region ions and electron motion the D-region is mainly driven by the neutral winds because the collision frequency is higher than the gyro-frequency ($< 80\,\mathrm{km}$). Therefore, the variability in the D-region is not only driven by solar radiation but is also significantly affected by neutral atmospheric dynamics. For example, the so-called D-region winter anomaly (i.e. enhancement in the electron density during the winter) is proposed to be caused by the atmosphere/ionosphere coupling process through the planetary wave, temperature, and composition changes by the vertical and meridional transport. Changes in the D-region electron concentration hinder the long-distance propagation of



high-frequency (HF) radio waves through absorption. Thus, understanding and quantification of the D-region electron density
variability and its causative mechanisms are essential.

Continuous observation of the D-region is very challenging because of the low electron abundance. There are very few
direct observational techniques such as sounding rockets (Mechtly, 1974), incoherent scatter radar (Chau and Woodman, 2005;
Baumann et al., 2022), VLF (Worthington and Cohen, 2021), and partial reflection radars or MF radars (Belrose, 1970; Igarashi

et al., 2000; Singer et al., 2008) that are effectively used to measure the D-region electron density (see e.g. Friedrich and Rapp,
2009, for a review). Among them, the rocket-borne in-situ measurements (Faraday experiment) are highly accurate (capable
to detect even low electron density) but the data availability is comparably sparse because the launching costs are high. Using
the available EISCAT radar data and rocket measurements Friedrich et al. (2004) present the quiet auroral ionosphere and its
neutral background. Particularly, for the representation of the D-region a limited number of rocket measurements were used.

In addition to the direct measurements, indirect information of the middle atmosphere ionization state by proxies like ri-
ometer absorption, measuring the reduction of incident galactic radio noise to a quiet day curve (Friedrich et al., 2002). The
quiet day curve is typically derived from the previous interference-free days (see e.g. Moro et al., 2012; Renkwitz et al., 2011).
The observed absorption is an integrative parameter attributed to the electron density at altitudes between 80-90 km due to
collisions between free electrons and neutrals. During polar cap absorption, the 55-ion Sodankyla model was applied in the

inversion of the raw densities measured by the ultra-high frequency (UHF) EISCAT radar, and that provided some realistic
estimates of the actual electron density in the D-region (del Pozo et al., 1999). Recently, long-distance sub-ionospheric very
low-frequency radio wave observation was also used as an indirect method to investigate the long-term climate change in the
D-region (Clilverd et al., 2017).

Apart from the observations, based on the available datasets and Chapman's theory, there are some statistical (McNa-

mara, 1979), empirical (Friedrich and Torkar, 1992; McKinnell and Friedrich, 2007; Friedrich et al., 2018a), semi-empirical
(Friedrich and Torkar, 2001), physical, chemical and theoretical models (Burns et al., 1991; Verronen et al., 2016; Zhu et al.,
2023) also exist in the literature, and they can reproduce the quiet time lower ionosphere reasonably well. For example, incor-
porating several hundred rocket profiles, Friedrich and Torkar (2001); Friedrich et al. (2018a) developed an empirical model
called the Faraday International Reference Ionosphere (FIRI) model. IMAZ is a neural network-based empirical model for

the lower ionosphere in the auroral zone (McKinnell and Friedrich, 2007) develop for the International Reference Ionosphere
(IRI) global model community. Combining two chemical models namely, the Mitra-Rowe simplified 6-ion model and a 35-ion
model developed at the Sodankylä Geophysical Observatory Burns et al. (1991) developed a D- and E-region chemical model
called Sodankylä ion and neutral chemistry (SIC) model. Later on, Verronen et al. (2005) pointed out that the SIC model
could correctly estimate the ionization and electron densities during solar proton events (SPE) in October-November 2003. A

comparative study of electron densities observed from the VLF and rocket measurement with the OASIS (Originally Austrian
Study of the IonoSphere) model showed that below 68-70 km the model data agrees well with observations. However, above
68-70 km the OASIS model fails to reproduce the observation (Siskind et al., 2018). Overall, the models can be used as a
climatological mean representative of the D-region, when it comes to the observational dynamical changes their reliability is
questionable at least below 80 km because of the limited availability of the D-region observational data.



As mentioned earlier, the diurnal and seasonal variation of the D-region electron density is primarily caused by the variability in solar zenith angle and the Lyman Alpha radiation, respectively. In addition to that, it also shows an asymmetry behavior that the solar variations could not explain. For example, recently Baumann et al. (2022) reported the electron density is higher during the sunset than the sunrise. The postulate that the change in the recombination rates is a plausible reason for this observed asymmetry. Similarly, earlier investigations showed that enhancement in the D-region electron density during the winter (winter anomaly, Offermann, 1979a, b). In winter's absence or reduction of solar radiation, solar EPP could be a primary cause of the observed winter anomaly. However, the winter anomaly is also observed in the absence of EPP, these are called the 'meteorological types' of winter anomaly (Offermann, 1980). There are several mechanisms proposed for the formation of the 'meteorological types' of winter anomaly, changes in the NO due to the transport process associated with atmospheric planetary waves, and sudden stratospheric warming events (Kawahira, 1985). Though there are several case studies and periodic campaigns carried out to explore the D-region electron density variations and their causative mechanisms, we are still lacking to address the seasonal variation of D-region studies using continuous observations.

A recent study by Macotela et al. (2021) reported that during the fall equinox, there is an increase in the noontime mean amplitude of VLF radio waves in high and mid-latitudes, which do not resemble the solar zenith angle (see Fig. 3). They called this a fall effect. They showed that during that time a decrease in the background temperature and an increase in the semi-diurnal tide (S2) amplitude is observed. Thus, they postulate that the changes in the collision frequency associated with the background temperature could be a prime reason for the observed increase in the noontime VLF amplitude. It is well known that the VLF radio waves are drastically affected by the D-region electron density, however, Macotela et al. (2021) did not consider the electron density role (if any) on the fall effect. Therefore, this study focuses to explore the altitudinal, and seasonal variation of noontime electron number density using partial reflection radar observations from 2014 to 2022 at a high-latitude location.

In the following sections, we will introduce the instrument used in this study, followed by a brief description of the experiment and methods to derive electron number densities in Section 2. This includes the advances we recently made in the processing and outlier removal by e.g. applying the FIRI model as a quiet reference. This also includes the detection of EPP and separates the radar data into quiet and solar and geomagnetic active periods. In section 3 the results, namely the annual noontime electron number density profiles including the nine-year climatology are presented. Furthermore, we discuss the specific features found and the probable connection to other phenomena. Finally, we conclude our findings in Section 4.

## 1.1 The Saura radar

The Saura radar is a modular partial reflection radar built in 2002 on the island of Andøya (69°N, 16°E). The radar is operated at 3.17 MHz, and due to the proximity to the medium frequencies it is often referred as an MF radar. The specialty of this instrument is the comparably large Mills Cross antenna array spreading over roughly 1x1 km (see Fig. 1). Each dipole antenna forming the sketched crossed dipoles is connected to an individual phase-controlled transceiver module capable of 2 kW peak



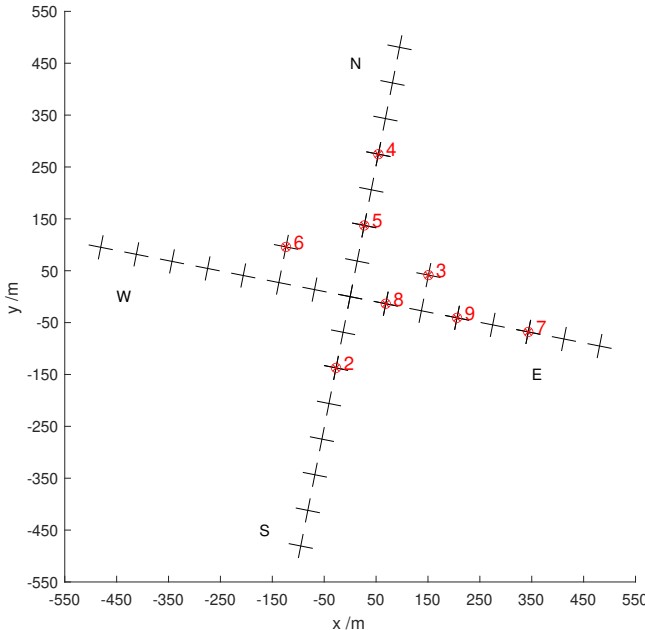

**Figure 1.** Sketch of the Saura radar antenna array, marked antennas are connected to individual receiver channels used for interferometry.

power. This configuration allows for the emission of circularly polarized pulsed radio waves and beamforming for different pointing directions, which makes it unique for this frequency range.

95 One major scientific target of this instrument is the measurement of winds in the lower ionosphere, namely the D-region. Such measurements are widely performed by a Doppler Beam Swinging (DBS) experiment, measuring the radial velocity components for distinct vertical and four oblique beam-pointing directions. The horizontal resolution for DBS is typically less than 10 km at 80 km altitude.

 Alternatively, the Full Correlation Analysis (FCA, e.g. Briggs, 1984) can be applied and is especially advantageous for
100 higher altitudes Imaging Doppler Interferometry (IDI, e.g. Palmer et al., 1995; Roper and Brosnahan, 1997), see Renkwitz et al. (2018) for more details and its application with the Saura radar. For the last years the Saura radar wind measurements are performed with a resolution of at best 4 min and the same cadence is applicable for electron density measurements. However, often larger windows of up to 12min are used to reduce the uncertainty.

 The latest upgrades to the radar included pulse code capability and the addition of three digital receiver channels to improve
105 the Signal-to-Noise-ratio and strengthen interferometric methods, respectively.



## 2 Electron density experiment description

Two methods may be used with sufficiently large MF radars to deduce electron densities in the D-region making use of observing the wave absorption and Faraday rotation. They are normally referred as differential phase and differential amplitude experiments (DPE, DAE), analyzing the radar echoes from alternating transmission of both magneto-ionic modes. These tech-

niques were described by e.g. Sen and Wyller (1960); Belrose (1970); Budden (1983), more recently by Vuthaluru (2003) and were also applied by e.g. Grant et al. (2004); Osepian et al. (2008) or Liu et al. (2020); Zhu et al. (2023) to the MAI and Kunming radar systems.

Equation 1 describes the relation of the amplitude ratio $A_x/A_o$ to the ratio of reflection coefficients $R_x/R_o$ for the height interval $\Delta h$ for DAE, in which $k_x$ and $k_o$ are the absorption indices and are related to the imaginary part of the complex

refractive indices by a factor $\omega/c$ for both magneto-ionic modes. For DPE the differential phases of the two modes and the reflection coefficients are compared and normalized by the real part of the refractive indices, $\mu_x$ and $\mu_o$ per height interval (Eq. 2)

$$N_{DAE}(h) = \frac{\Delta(ln\,R_x/R_o) - \Delta(ln\,A_x/A_o)}{2(k_x - k_o)\Delta h} \qquad (1)$$

$$N_{DPE}(h) = \frac{\Delta(\phi R_x - \phi R_o) - \Delta(\phi A_x - \phi A_o)}{2(\mu_x - \mu_o)\Delta h} \qquad (2)$$

For this study, we restrict to DPE where we applied the high-latitude approximation of the Sen Wyller refractive index after Flood (1980). For the Saura radar, equivalent experiments have been carried out by Singer et al. (2008, 2011); Renkwitz et al. (2021), but also e.g. with the Buckland Park radar by Holdsworth et al. (2002). The pressure and temperature profiles are taken from CIRA and for the magnetic field, 51469e-9 T after IGRF and $\theta = 12.2°$ are used for the radar location.

Electron densities have been derived from two different experiments probing the atmosphere with alternating circular polarization matching both magneto-ionic components (ordinary, extraordinary). The vertox experiment is an uncoded vertical-only

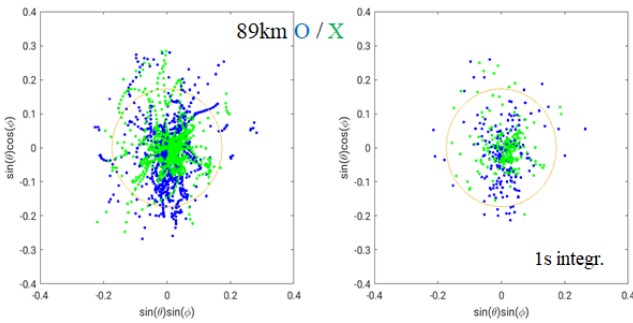

**Figure 2.** Example of derived AOA positions for O- and X-mode depicted in blue and green, respectively, left: original resolution and right: with 1 s integration time. The orange circle marks the selected $10°$ area of interest.



**Table 1.** Saura experiment parameters.

| Saura | vert-ox | sd-07ox-4c |
|---|---|---|
| beam directions | $\theta = 0°, \phi = 0°$ | $\theta = 0°, \phi = 0°$ |
| | | $\theta = 6.7° : \phi = 56.2,$ |
| | | $146.2, 236.2, 326.2°$ |
| code | single pulse | 4bit compl. |
| range resolution | 1000 m | 1000 m |
| pulse repetition | 75/30 Hz | 100/30 Hz |
| run time | 215 s | 235 s |

experiment, whereas the 5dbs-ox-4c is a 4-bit complementary code Doppler Beam Swinging experiment (DBS) probing five
different directions primarily intended for wind measurements (see Table 1 for more experiment details). The experiments
have a duration of 215 s and 235 s for the vertical-only and DBS, respectively, which by that covers multiples of the typical
correlation time of the observed structures. The shorter runtime of the vertical experiment is chosen to allow a short high-range
monitor experiment, which is not used here. Thus for most of the measurements, almost continuous sampling is maintained
without larger gaps. The lesser number of data points in the DBS experiment per beam pointing direction is partly compen-
sated by the 4-bit complementary code. This pulse coding improves the signal-to-noise ratio as the averaged transmitted power
increases, as well as received power from random noise and interference, are reduced as they do not match the code. After de-
coding the multi-beam DBS experiment we only analyze the vertical measurements for electron density estimations. Equivalent
electron density measurements with the Saura radar have been lately also used to investigate e.g. so-called polar mesosphere
winter echoes to study their occurrence (PMWE, Renkwitz et al., 2021). Furthermore, Saura electron density profiles have
been partly validated by rocket-borne in situ measurements during the PMWE campaign (Strelnikov et al., 2021; Staszak et al.,
2021).

**2.1   Amendment of the electron density estimates**

The typically observed rather large variability of electron density profiles gives rise to multiple potential sources within the
probed volume. Assuming a $6°$ wide transmit beam, the illuminated area corresponds to a diameter of $\approx 8.5$ km at 80 km
altitude, neglecting side lobes. The horizontal variability of gradients in the electron density for this size is likely, caused
by the presence and superposition of propagating small-scale waves. Since the earlier electron density measurements with
the Saura radar (Singer et al., 2011), the fundamental technique is combined with additional signal processing methods and
interferometry.

First, we split the complex times series into 4 blocks, then the correlation length of the detected echoes for each block and
range is derived. The correlation length is used to apply a sufficient amount of coherent integration to reduce the variability of
the raw data in order to stabilize the angle-of-arrival (AOA) estimates.



To improve the individual electron density estimates for each experiment run, we remove radar echoes that are received most likely through the side lobes of the antenna array. Given the sparse character of the Mills-Cross-like antenna layout, imperfections in the radiation pattern, namely side lobes along the axes of the antenna array exit. For situations when much larger electron density gradients exist in the orientation of these side lobes rather than at the nominal main beam pointing direction, echoes from these unwanted directions are received and will impair the estimates.

Therefore, we apply interferometric methods to discriminate the location of the scattering structures. For localizing the scattering structures we generally make use of four individual antennas, marked in Fig. 1 as receiver channels 3-6. With this arrangement, six individual and non-redundant baselines can be used for AOA estimates covering up to $25°$ given its shortest baselines of $1.48\lambda$ (see e.g. Renkwitz et al., 2018; Renkwitz and Latteck, 2019). The same arrangement and basic methodology were already applied using IDI to make use of the ionospheric inhomogeneity for Saura wind measurements (see Renkwitz et al., 2018). The required phase information of the receiving channels is estimated based on AOA statistics for all beam positions for the course of months.

An example of derived instantaneous AOA positions for a single vertical sounding experiment of less than 4 min runtime is shown in Fig. 2 for a range of 89 km from the radar. Two scenarios are shown, the native resolution and after applying an integration of 1s before solving for AOA. From that plot, it is obvious that still many positions are outside the marked $10°$ area of interest. Furthermore, as the radar echoes are measured in range gates, basically a position on the spherical shell, it will result in vertical smearing without proper conversion and selection. For each of the four time series blocks, the AOA positions are used to potentially discard the derived differential phase between the ordinary and extraordinary wave for the same chunks.

Another important point is a suitable unwrapping of the differential phases. Even though a progressively increasing radial phase difference is expected with increasing electron density over altitude, we also allow for minor negative estimated phases to cope with measurement errors. Larger negative phases, however, are interpreted as positive phase wraps. Note, a reduction of electron density may occur for electron bite-out situations during the charging of ice particles in the summer mesosphere (see PMSE, Rapp et al., 2003; Rapp and Lübken, 2004; Friedrich et al., 2011).

The first cleaning of the data is done by removing isolated data points, like potential erroneous data at the lowermost captured altitudes. Additionally, outliers based on the typically observed variability of values adjacent in time and altitude are flagged.

In the next step, we aim to reject suspicious electron density profiles, which show excessively small or large values. For this, we use FIRI (Friedrich et al., 2018b) profiles that are selected for the nearest latitude ($60°$) as well as solar flux and then interpolated for DOY and solar zenith angle (SZA, see Fig. 3). The interpolated FIRI local noontime electron densities for low solar flux conditions (sfu=90) are depicted in Figure 4. We are aware FIRI is not explicitly meant to be representative of polar latitudes, but we apply the profiles as a coarse reference for geomagnetic quiet times.

For the evaluation of the Saura electron density profiles for quiet conditions, we allow for deviations between 1/10 and four times to the reference profile. The same methodology is used for geomagnetically disturbed times, but we allow for much larger densities (100 times the FIRI profile). For the main target of this study, namely deriving daily noon averages, we average the remaining profiles for both scenarios between 9-13 UT.





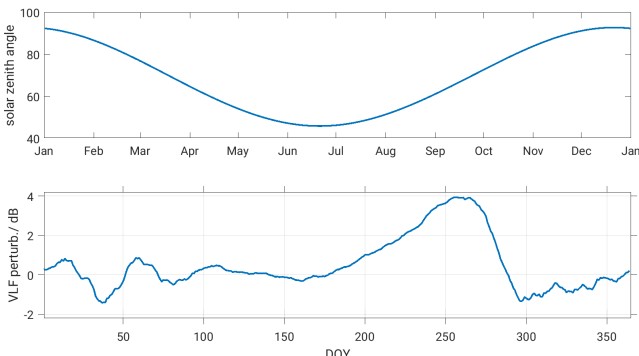

**Figure 3.** Top: Daily noon solar zenith angle for the Saura location. Bottom: Detected mean VLF amplitude perturbations for NAA-SOD after Macotela et al. (2021).

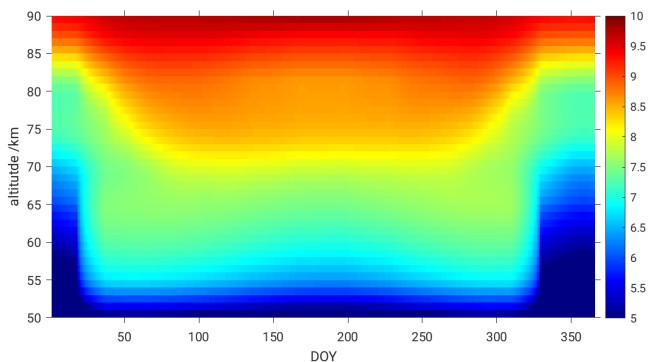

**Figure 4.** Electron densities as derived from FIRI for $60°$N during low solar flux conditions (sfu=90), interpolated for the SZA at noontime DOY. Colorbar represents densities as $log_{10}(ne)\,m^{-3}$.

## 2.2 Discrimination of quiet and disturbed conditions

As already mentioned, enhanced solar and geomagnetic activity will alter the current state of the lowermost ionospheric layer until it settles down and recombination is completed. For our purpose to derive climatologies for quiet, but also for disturbed conditions, these different conditions need to be separated. One obvious and frequently used way to evaluate the geomagnetic condition is to use proxies derived from e.g. magnetometer data like k, Hp, etc. Often, these parameters are then analyzed in combination for multiple locations. Depending on the distribution of these locations the proxies might be more relevant for

mid-latitudes and as planetary indices. Additionally to this and solar radiation like f10- and f30-flux and solar wind parameters are useful to describe the current influx. Particle detectors onboard GOES/POES satellites and comparable instrumentations may also discriminate for certain energy bands of precipitating particles.



Fortunately, the Saura radar itself is very sensitive to electron density enhancements occurring in the radar's field-of-view, see Renkwitz and Latteck (2017). Significantly enhanced electron densities, e.g. caused by EPP, form virtually isolated layers

in the echo power profiles as the radar signal gets strongly absorbed for the altitudes above. First observations of such virtual layers were reported and named ILME (Isolated Lower Mesospheric Echoes, Hall et al., 2006) and were widely attributed to events of precipitating protons during solar flares. The time series of such isolated layers for the years 2003 to 2022 as detected by Saura (extended data set as Renkwitz and Latteck, 2017) is used in the following to flag disturbed conditions and treat the electron density data for both scenarios separately.

Based on the ILME detections using the Saura radar, periods of EPP events are flagged and thus excluded from further processing of the quiet periods' profiles. To prevent contamination from undetected events and to cover the recombination time, we use a window of $\pm 1$ hr around the individual EPP detections. In the next step, the daily noon medians for all data and quiet times only are calculated for each altitude from all valid measurements during the period of 9-13 UT, and stored for the corresponding DOY.

An example of a successful differentiation for both disturbed and quiet ionospheric conditions and corresponding annual noon electron densities are shown for the year 2019 in Fig. 5. In the upper panel, all valid data including active periods (e.g. EPP and SPE) are shown, where enhanced electron densities are clearly visible around March-April and September-October for altitudes of 60-70 km. Such enhancements near the equinoxes are commonly related to the Russel-McPherron-cycle (Russell and McPherron, 1973). The visible vertically aligned gaps are caused by the rejection of long-lasting EPP events rather than

observational gaps. The bottom panel depicts the quiet time (excluding EPP and SPE) D-region electron densities during the year 2019. Interestingly, even after removing the geomagnetic disturbances effect, the enhancement in the electron density during the spring and fall equinox is prominent, the probable cause and effect of these enhancements will be discussed in the following.

## 3  Analysis and discussion

The composite mean of nine years (2014-2022) of electron densities during geomagnetically disturbed and quiet conditions are provided in Fig. 6 top and bottom panels, respectively. The general picture of the electron number densities in the lower ionosphere is governed by the dominating incident solar radiation. As an indicator of incident radiation the flux of UV (Lyman-alpha, and X-rays in the case of solar flares), but also the solar zenith angle (SZA) is used. Assuming a rather constant flux and a symmetrical SZA for spring and autumn (see Fig. 3) also a symmetric electron density is expected, neglecting other

contributing factors like dynamics. This assumed symmetry of spring and autumn electron densities is not observed in our data, but increasing D-region electron densities below 75 km altitude from spring towards autumn as shown for 2019 in Fig. 5.

It has to be noted, that 2019 was characterized by generally extremely low solar and geomagnetic activity and might therefore represent an extreme case. Therefore we also applied the same methods to years of enhanced solar activity, from 2014 (the maximum of solar cycle 24) until the end of 2022 (onset of solar cycle 25). The analysis covers a total of nine consecutive

years, since the last major radar upgrade, spanning from a rather poor solar maximum to extreme solar minimum conditions,



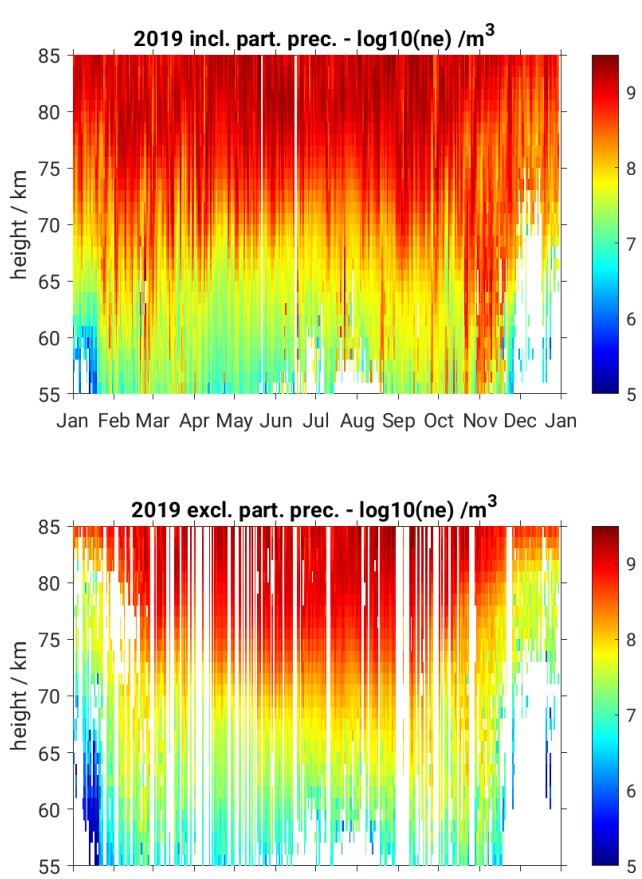

**Figure 5.** Observed Saura electron densities during 2019, for all conditions (top) and excluding active periods.

resembled by monthly smoothed solar sunspot numbers of 130 and down to 0, and corresponding solar 10.7cm flux of 150 to 67 SFU, respectively (see NOAA, 2023).

For all data of each year, the 2-hourly solar flux is used to derive FIRI profiles that are then employed to reject obvious outliers from the radar measurements. For a climatological picture, we calculated median electron densities for each DOY over all years, which is shown for both conditions in Fig. 6. Even though we are averaging over fairly different solar and geomagnetic conditions, the general asymmetry which was already found for the year 2019 prevails also for this climatology. For a constant number electron density of $100/m^3$, an almost linear slope from the beginning of April to the end of September is visible for 72 to 66 km, respectively. Such a slope was already visible for the year 2019 data shown in Fig. 5.

Noteworthy, in February and March as well as in September and November enhanced electron densities are visible below 65 km altitude. One plausible cause for these enhancements might be undetected EPP events creating a bias in the shown medians. However, enhancements are also seen in the FIRI output for altitudes as low as 55 km from April to November (see





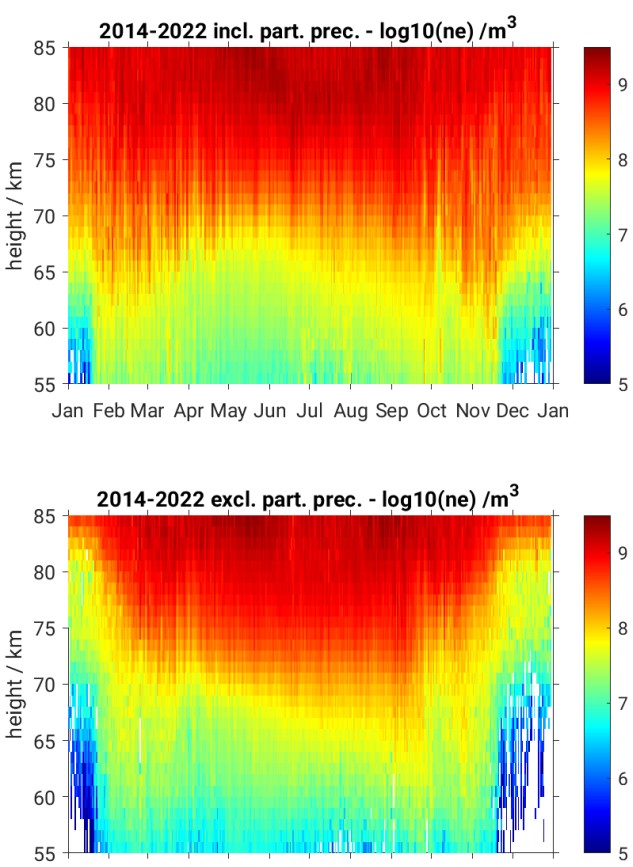

**Figure 6.** Climatology of electron density including (top panel) and excluding particle precipitation events for the years 2014-2022.

Fig. 4). Furthermore, there might be dynamics-driven causes for the observed variability as it's a rather unstable period with its proximity to sudden stratospheric warmings (SSWs) during the winter.

Note, the abrupt changes at the end of January and November appear artificial and are likely caused by the FIRI reference
applied to reject too large values of the measurements. The mentioned period actually describes the time of lowest SZA, when the sun is basically below the horizon every day, and represents the most challenging scenario to derive reliable electron densities. One obvious reason is the generally very low ionization and steep gradient near 85 km altitude (cf. Fig. 4), which doesn't allow for much range for radar measurements having in mind a minimum range resolution of 1 km. The upper panel of Fig. 6 depicts the median of all valid data including EPP events. Enhancements around the equinoxes are visible and
especially during the winter period. Even though the frequency of strong and long-lasting events is rather low in wintertime, their contribution may still be significant as they typically occur near noontime.



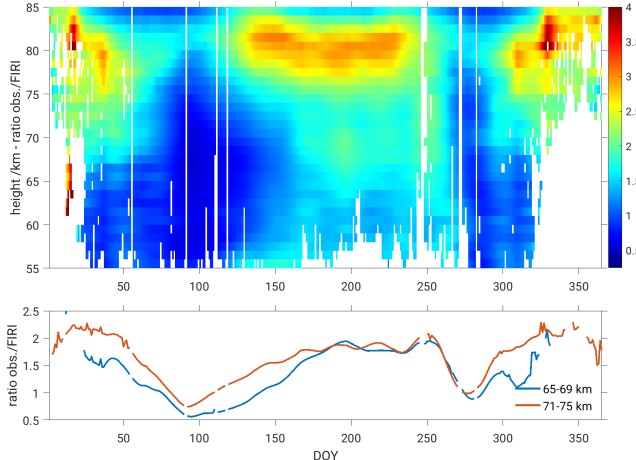

**Figure 7.** Linear ratio of observed median electron densities after 30 day smoothing and FIRI. Bottom: Median ratio for the altitude bins 65-69 km and 71-75 km.

The ratio of observed quiet time median electron densities smoothed with a 30-day window, and the very SZA-symmetric FIRI is shown in Fig. 7. Data covered by less than four years of quiet time observations are blanked as they may not be representative. For altitudes above 75 km the pattern appears symmetric to the SZA Specifically interesting are the altitudes near 70 km, which show enhanced densities from June to the mid of September.

As we initially highlighted the focus of this study is on the asymmetry between spring and autumn as previous studies indicated such behavior in the detected VLF amplitudes of long-distance transmissions (Macotela et al., 2021, see bottom panel Fig. 3). This asymmetry of spring and autumn is also clearly visible in the electron density measurements as depicted in the bottom panel of Fig. 7.

Generally, the period from the end of summer and mid of fall is a rather dynamical time of the year, for which we will raise some presumably related observations. Macotela et al. (2021) argued that changes in the temperature and semi-diurnal tidal amplitude (see Conte et al., 2018) could be a cause for the fall effect. However, their study did not include electron density's role in the observed fall effect (if any). It should be noted that the VLF amplitude is highly influenced by the electron density gradient in the D-region ionosphere (Silber and Price, 2017). Thus, the present results suggest that in addition to the dynamical effects, enhancement in the electron density also could contribute to the observed fall effect in the VLF amplitude. Iimura et al. (2021) investigated the quasi-2-day wave structure (Q2DW) for multiple latitudes using ground-based and satellite measurements. Noteworthy, they found enhanced Q2DW activity during summertime between 75-95 km altitude for 60°N. This enhancement fades towards the zonal wind reversal (DOY 260), which is also coincident with the decrease of the semi-diurnal tides. Another interesting result also noted in our electron density profiles is at the beginning of October, a rather sharp and sudden decrease in electron densities is visible, which can be related to the so-called October effect (Pancheva and Mukhtarov, 1996). A challenging aspect of the present result is what could be the source of these electron density enhancements (reduction)




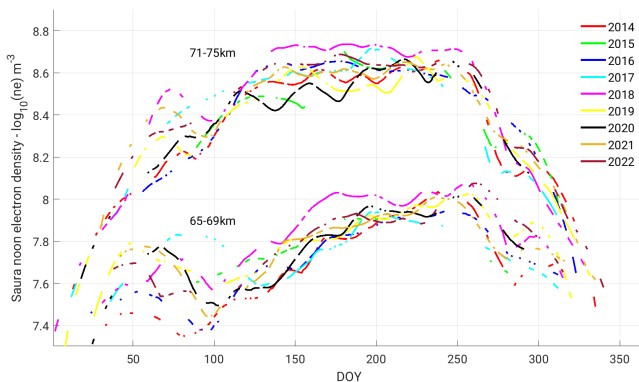

**Figure 8.** Averaged electron densities for the altitude bins 65-69 km and 71-75 km including a 30 day smoothing.

during the fall (October) condition. One of our current understandings and hypotheses is that during the late summer and autumn period downwelling occurs at mesospheric altitudes as a part of mean meridional circulations. Such downwelling will transport more NO from the lower thermosphere to the observed altitudes and planetary wave circulations transport NO-rich air equatorward to mid-latitudes, where it is ionized to produce high electron densities through ionization (Garcia et al., 1987).
If the meridional circulation associated with downwelling is the cause of the enhanced electron densities, then one should observe the high electron densities in October as well. Thus, it gives a hint that there could be some other process also involved in the noted changes in the electron densities during the fall condition. Coincidentally, Conte et al. (2018) reported a decrease in the S2-tidal amplitude during October for middle and high latitudes. During this period, a very pronounced and consistent feature occurs in the mesosphere, namely the zonal wind reversal (see e.g. Keuer et al., 2007; Hoffmann et al., 2010; Jaen et al.,
2022). This rises the question of whether the changes in the S2-tidal amplitude and zonal wind reversal have any impact on the D-region electron density, and if so how?

For a better inter-annual analysis, the median electron densities for 65-69 km and 71-75 km over DOY are shown separately for each year in Fig. 8 after smoothing with a 30-day window. The annual asymmetry is seen for both altitude bins but very
pronounced for the lower altitudes, where an increase by a factor of at least three is seen from day 100 to 250. The sudden decrease of electron density occurs around day 265 but differs by about 10 days between the higher and lower altitude bins. For both, the exact timing of the increase around day 150, as well as the decrease around day 250, shows comparably little variability seen throughout the years for the higher altitude window. Whereas it seems less stable for the lower altitudes, with spreads of 20 days, giving rise to potentially larger sensitivity to other phenomena like dynamics. An important point here is
that the fall effect is present in all the years with minor year-to-year variations in the onset time, irrespective of solar activity.

Despite the general increase for the lower altitudes from DOY 100 to 250 for most of the years, especially for 2018, an early and rampant behavior until DOY 170 and a rather continuous noon electron density level until DOY 270 is seen. A similar peculiarity is prominent for the upper altitude bin. Given 2018 was a year with extremely low solar activity the found annual development of electron densities seems rather surprising. The basically contrary is seen for 2014, near the solar maximum of



cycle 24, where low electron densities in spring and a steady increase until DOY 250 are observed. The tendency for somewhat larger electron densities below 70 km altitude during low solar flux, 70 SFU for solar minimum and 150 SFU for 2014 solar maximum, is also visible in FIRI, while it is the contrary for the altitudes above. In the previous study of Macotela et al. (2021) the majority of the year 2018 did not show any significant deviation from other years. However, slightly enhanced VLF amplitudes were seen for the second week of October. Noteworthy, these detected amplitudes are derived for the propagation path between most northeastern USA (Cutler, ME) and northern Finland (Sodankylä). Thus, the electron density profiles of the entire path need to be taken into consideration, and the local Saura profile might have only little influence. Thus, more local phenomena could have caused the observed higher electron densities in 2018. However, these need further investigation to explore the exact reason.

## 4   Conclusions and outlook

In this study, we present local noontime climatologies of electron number density in the middle atmosphere at high latitudes for solar and geomagnetically quiet periods, but also including disturbed periods. The data was derived by active high-frequency radar experiments exploiting the phase information of radar echoes, that correspond to Faraday rotation caused by electron number density in Earth's magnetic field. To improve the long-time established methods we applied further signal processing as well as interferometry techniques. The latter is used to reject radar echoes from far off the beam-pointing direction. We furthermore involved FIRI model output to exclude obvious outliers from further processing, which proved to be very useful for SZA below 90. Geomagnetically active periods including EPP events as well as SPE were detected with the radar and these periods were treated separately from the quiet times. Finally, local noontime electron density profiles were derived for both situations.

We found a clear asymmetry between spring and autumn, which is not explainable by the SZA nor it is visible in the ionospheric models. However, recent VLF experiments have shown a comparable asymmetry in detected VLF amplitudes. Furthermore, we found a consistent and steep decrease in electron densities around DOY 265.

Our current understanding is that downwelling is associated with the meridional circulation during the fall and winter seasons and could be responsible for the enhanced electron density through the transport of NO from the lower thermosphere, however, this mechanism could not explain the reduction in the electron density during the first week of October. In addition, during the late summer and autumn periods a temperature reduction and an increase in semidiurnal tide could also have a significant role in the increase/decrease in the electron density.

The pattern of asymmetric spring-autumn and a steep decrease in October was found for all nine years of data analyzed in this study, while the interannual variability is restricted to a few days, which might be surveyed in more detail at a later point.

We plan to investigate the found peculiarities like the enhanced densities of 2018 in more depth in a successive study, where we aim to incorporate both methods, absorption, and phase rotation measurements and incorporate additional modeling efforts. Furthermore, we plan to extend the measurements to the technologically rather similar, but smaller radar system in Juliusruh



at middle latitudes, which in the past was mostly used for the study of dynamics (see e.g. Hoffmann et al., 2010; Jaen et al., 2022).

## Author contribution

325 TR had the main responsibility of the radar experiment, its data analysis and writing of the article. SM and JJ helped to developed the concept and interpretation of data and discussion. WS contributed to the fundamental electron density analysis and discussions of results. All authors have read, corrected and agreed to the submitted version of the manuscript.

## Data availability

The data needed to reproduce the figures including the observations is shared through radar-service.eu. Files are MATLAB
330 data files. Temporary link to the data:

https://www.radar-service.eu/radar/en/dataset/

CgMgGNtKxaiRnjGR?token=GLVhWXxvewMEbCvifjhp

The data will be available after publication through registered DOI: 10.22000/993

## 335 Conflict of interest

The authors declare no conflict of interest.

## Acknowledgments

This research has been supported by "AMELIE - Analysis of the MEsosphere and Lower Ionosphere fall Effect" (DLR project D/921/67286532) as well as by the Deutsche Forschungsgemeinschaft (VACILT, grant PO 2341/2-1) and Bundesministerium
340 für Bildung und Forschung grant 01 LG 1902A, TIMA in the frame of ROMIC. We appreciate suggestions by Jorge L. Chau and the support by the Andoya Space Center for Saura radar operations.



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
