# Peer review of "Ground-based noontime D-region electron density climatology over northern Norway"

_EGUsphere, 2023_

## Author Response (AR1)

Renkwitz et al., Ground-based noontime D-region electron density climatology over northern Norway

Submitted to ACP, 2023 – renkwitz@iap-kborn.de

The answers have already been published in the discussion on the ACP website.

Answer to Rev #1:

The manuscript "Ground-based noontime  ..." by Renkwitz et al. presents an interesting new source of continuous D-region measurements. After an overview of the processes active in the D-region, a description of the updated Saura radar is given. Data taken at local noon at Andøya of nine years of observations are presented. Since at high latitudes (69°) additional ionisation by energetic particles is very common, the quiet data were filtered out using the (non-auroral) FIRI model as a coarse reference to identify disturbed conditions. One aspect of the analysis was to find the so-called Fall Effect,  an electron density enhancement typically ocurring in October, which could indeed be found at this relatively high latitude. The paper is clearly arranged, but in some places I took the liberty to suggest other wordings. Beyond these "peanuts" I have the following comments:

lines 123/124: I am surprised that the neutral atmosphere was taken from CIRA (which year?), notably since the authors are from the same institution as F.J. Lübken and U. v. Zahn, the pioneers in establishing a local atmospheric model based on met-rocket soundings from Andøya.

lines 244/245: "Enhancements around the equinoxes are visible and especially during the winter period". Equinox occurs when day and night have the same length; hence equinox occurs typically near March 20 and September 29 (but not in winter)

Citation: https://doi.org/10.5194/egusphere-2023-815-RC1

We appreciate the helpful comments from Reviewer #1, Martin Friedrich.

1)

As a reference for the neutral atmosphere we used CIRA version 1986, which we unfortunately missed to quote precisely and did not reason it.

Indeed, there is a climatology by Lübken, JGR, 1999 followed by Lübken and Müllemann, JGR, 2002, derived from falling spheres at Andenes.

One limitation of this climatology for our purpose is, it only covers the summer period, namely DOY 113 to 266. There also have been a few flights during winter, but for our study we require a consistent reference throughout the year.

There are indeed some noticeable differences between CIRA-1986 and Lübken-1999, which are rather intense above 85 km altitude thoughout the year and comparably small below 60km for DOY 170-200 and 70 to 80km towards the winter.

Lübken 1999 summarized the deviations to CIRA-1986 to be within about +/-5% for altitudes below 80 km.

As sketched in the outlook we're also working on applying the same technique to the widely comparable Juliusruh radar (54°N), for which no rocketborne climatology exist.

As we only show electron densities below 85 km and generally handle measurements with the Saura radar above this altitude with care, we tend to accept the mentioned uncertainties to be tolerable in the advantage of a homogeneous and consistent reference.

We'll add a corresponding comment to the manuscript, which is certainly needed.

2)

We realize the formulation of this rather short sentences is misleading.

We wanted to express that enhancements are visible around the equinoxes to which EPP (Russel-McPherron-cycle) are typically attributed, but enhanced electron densities are also seen for the winter period.

We'll rephrase this statement.

Answer to Rev #2:

We are grateful for the comments by Rev. #2 helping to improve the manuscript and its reasoning of the observed climatology.

General)

"...and a sharp decrease at the beginning of October, but fails to give insight into these statistical characteristics"

We are in doubt which statistical insights are desired. We derived daily local noontime electron densities for the years 2014-2022. As an example, the variability for 2019 is shown in Figure 5 for both scenarios with and without particle precipitation. The variability for the individual years for two altitude sections are given in Fig. 8 and discussed it near Line 280.

The observed daily noontime variability of electron densities is large at the lowermost altitudes given the potentially sparse data and partly significantly different solar zenith angle during the selected daily window. For the altitudes above 65km (70km) the ratio of daily noontime standard deviation over median electron density (a relative standard deviation) is mostly below 0.4 (0.3).

We herewith add a plot of the daily standard deviation for the composite shown in Figure 6.

[Figure]

Title)

We agree it's rather inconsistent to use "electron density" as well as "electron number density", which however is found frequently in an equivalent usage throughout the literature.

We prefer "electron density" and will adapt the manuscript accordingly.

Line 5)

We'll modify to: upper middle atmosphere (50-90km)

Line 6)

We'll add (2014-2022) here, but mentioned it in detail in Line 223, which might be rather late.

Line 7~13)

We'll rephrase these.

Line 10)

We'll add a note to Line 7~13 to highlight the gradual electron density increase towards September.

Line 65)

Unfortunately, we missed to explain EPP in the submitted version: energetic particle precipitation. We already added it.

Line 72) The relevance of our study is the unexpected asymmetry of electron density, which can not be explained by the symmetric solar zenith angle as the primary source of ionization. This asymmetry is e.g. not shown in models like FIRI.

The derivation of a collision frequency climatology would be very useful, however, we're actually using the CIRA model to estimate the collision frequency in order to derive electron density. The existing pressure or collision frequency climatologies do not show such large asymmetry. Please also see a corresponding answer to Reviewer #1.

Macotela et al. (2021) basically was just the initial point to look for a potential course for the observed VLF amplitude asymmetry as the VLF propagation paths investigated should be near 70km altitude. Within this publication a connection to mesospheric temperatures as well as changes in the tidal intensities was given.

Line 173)

The presence of PMSE indeed can reduce the number of free electrons when they are attached to ice particles (Friedrich et al., 2011) and thus might create a depletion during June/July at 85km altitude. However, we can't connect them to the gradual increase of electron density at altitudes below 70km, e.g. Fig. 7.

Besides the reasoning of Macotela et al. (2021), we speculate about the relevance of the meridional circulation and prevailing downwelling during autumn. Such dynamics can enhance electron densities towards October as more NO is transported into the observed altitudes.

To support this idea we added a corresponding climatology of the meridional winds for the same years as new Fig. 8. The pattern shows enhanced winds starting in October that might also be connected to the sudden decrease of electron density.

Line 178)

DOY = Day Of Year, we missed to add a corresponding note.

Line 180)

The local noon occurs at the radar location around 10:55UT. We selected 9-13UT to have multiple measurements per day to get a robust median value, potentially even for geomagnetically disturbed days.

Line 218)

true, we also added it to Line 75

How do the VLF experiments show consistency with the observations presented in this study?)

The observed VLF amplitudes (Macotela et al., 2021) are replicated in Fig. 3 bottom panel while a corresponding electron density graph is shown in Fig. 7 bottom panel.  We refrain from showing both together as our measurements represent the local property, while for the VLF it's an integral of the entire propagation path (northeastern USA and northern Finland). Still, the gradual increase between DOY 100/150-250 is clearly visible. A corresponding description is given from Line 251 on.

Please give a possible explanation or more discussion of the spring and autumn asymmetry.)

Our understanding is, that the conditions during spring are as expected, the electron density increases throughout the summer corresponding to the solar zenith angle changes.

However, during autumn the rather high electron densities are unexpected and need to be connected to dynamics e.g. meridional circulation and downwelling.

To support this, we added an additional figure showing the meridional wind climatology for the same years.